# ROAD: Learning an Implicit Recursive Octree Auto-Decoder to Efficiently Encode 3D Shapes

**Sergey Zakharov, Rareş Ambruş, Katherine Liu, Adrien Gaidon**

Toyota Research Institute

**Abstract:**

Compact and accurate representations of 3D shapes are central to many perception and robotics tasks. State-of-the-art learning-based methods can reconstruct single objects but scale poorly to large datasets. We present a novel recursive implicit representation to efficiently and accurately encode large datasets of complex 3D shapes by recursively traversing an implicit octree in latent space. Our implicit Recursive Octree Auto-Decoder (ROAD) learns a hierarchically structured latent space enabling state-of-the-art reconstruction results at a compression ratio above 99%. We also propose an efficient curriculum learning scheme that naturally exploits the coarse-to-fine properties of the underlying octree spatial representation. We explore the scaling law relating latent space dimension, dataset size, and reconstruction accuracy, showing that increasing the latent space dimension is enough to scale to large shape datasets. Finally, we show that our learned latent space encodes a coarse-to-fine hierarchical structure yielding reusable latents across different levels of details, and we provide qualitative evidence of generalization to novel shapes outside the training set.

**Keywords:** Implicit shape representations, Reconstruction, Data compression

## 1 Introduction

Accurately and efficiently representing 3D geometry is a cornerstone capability in computer vision and computer graphics, with many practical applications in robotics and artificial intelligence. Decades of research in this area have produced a myriad of approaches, from traditional explicit methods [1, 2, 3, 4, 5, 6, 7] to learning-based implicit representations that encode shapes in the weights of neural networks and use various learning cues such as Signed Distance Fields [8, 9, 10], occupancy [11, 5], or radiance [12, 13, 14, 15]. Choosing one representation over another typically involves various tradeoffs between accuracy, scalability and generalization [16, 17, 18]. Related methods have either focused on modeling single shapes with increasing levels of accuracy at higher costs in terms of memory or time [12, 19, 20, 21] or on modeling classes of shapes, typically with single MLPs, which results in the ability to generalize to novel shapes as well as adapt to test-time data via differentiability but at the expense of high-frequency details [8, 22, 23, 24].

In this paper we address these key challenges through a novel neural network capable of simultaneously encoding a large number of shapes to a higher level of accuracy than previously possible. We build on recent advances which use neural fields (i.e. neural network parameterizations of continuous functions defined on Euclidean space) to represent any topology with arbitrary precision [18] combined with continuous, generative latent spaces for 3D shape generation [8, 25]. We aim to recover a compact function which learns to map complex topologies to an implicit space of shapes, without sacrificing reconstruction accuracy or differentiability. Our key insight is based on the fact that the neural field paradigm maps spatial coordinates to an encoding of the surface and thus requires thousands of evaluations to extract the underlying surface; moreover, the underlying implicit function needs to accurately model space outside the target geometry [26, 27, 28]. Alternatively, some methods explicitly define the relationship between an underlying explicit data structure and the latent space [22, 20], thus partitioning the implicit function.

6th Conference on Robot Learning (CoRL 2022), Auckland, New Zealand.

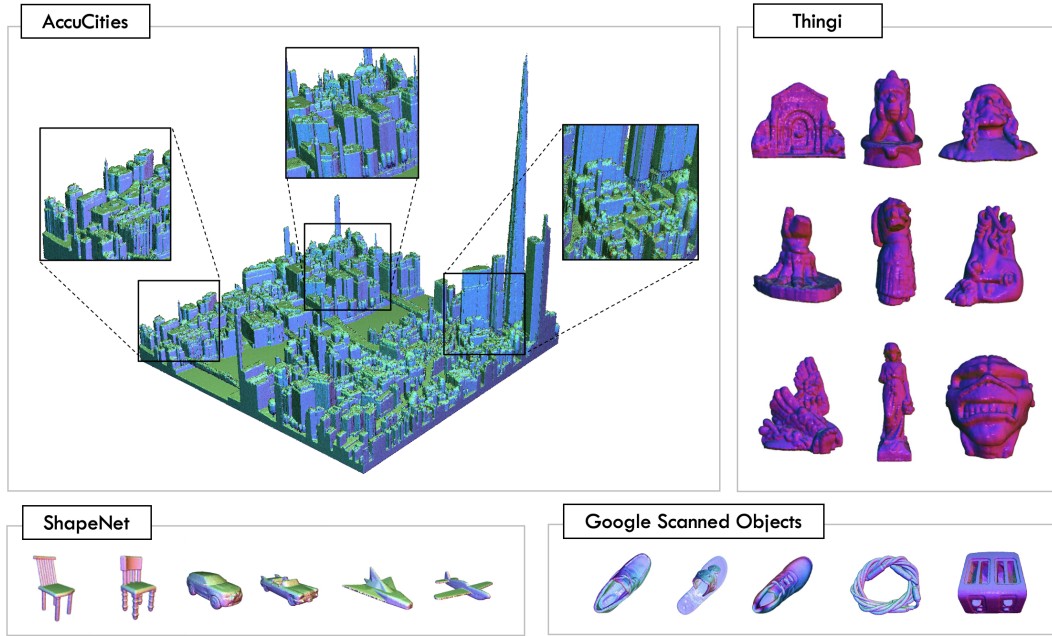

Figure 1: Our implicit Recursive Octree Auto-Decoder, **ROAD**, can represent many diverse shapes accurately and precisely with a small memory footprint thanks to recursive decoding of a hierarchically structured learned implicit shape representation.

To achieve high compression while still being able to reconstruct high-frequency details, we propose an implicit Recursive Octree Auto-Decoder (ROAD) formulation that operates entirely in the latent space and is guided by an octree partitioning of the space. The octree data structure provides a simple yet elegant solution for increasing surface detail while traversing down the tree [29, 20, 22], as well as an intuitive setup for a curriculum learning schedule where learning progresses according to a coarse-to-fine approach. Our method uses a single neural network to map a latent vector to eight other latent vectors corresponding to its eight children in 3-dimensional space; in turn, each of the predicted latent vectors can be fed back to the network for further subdivision. To extract a surface, we simply traverse down the tree starting from a single root latent vector, expanding nodes as needed based on predicted occupancy, until the desired level of resolution is reached. The output of a forward pass of our network is the actual surface, and it can be obtained in milliseconds, unlike seconds or minutes for related methods requiring complex operations.

Our formulation leads to a latent space that captures shape similarity and hierarchy in a manner that is conducive to high compression as well generalization to shapes outside of the training set. Our efficient implementation leads to a reduction of up to 99% in terms of storage space compared to the original mesh models; this includes network weights as well as any other learned features needed for reconstructing the meshes. Finally, we explore the relationship between 3D modeling power (as measured by the Chamfer distance), latent space dimension and dataset size, and show how to tune our model's capacity with a single hyperparameter — the dimension of the latent space.

Our contributions are summarized as follows:

- A novel implicit representation parameterized by a recursive function that encodes an arbitrary number of 3D shapes in a shared latent space, while retaining high reconstruction fidelity and requiring up to 99% less storage space compared to the input mesh models;

- A curriculum learning method that naturally exploits the octree spatial data structure through a coarse-to-fine optimization scheme;

- An analysis of the scaling law that correlates latent space dimension, dataset size and 3D reconstruction accuracy and a qualitative analysis of the learned latent space indicating a coarse-to-fine hierarchical structure resulting in reusable latents.

## 2 Related Work

**Neural Fields** are continuous coordinate-based neural networks that encode an underlying property of a scene. The popularity of these representations has increased dramatically as recent results have shown that with enough modeling power coordinate-based networks can be used to encode underlying physical quantities with arbitrary levels of precision [18, 8, 12]. Applications of these techniques include modeling 3D shape [11, 5, 8, 9], appearance / radiance [12, 13, 14, 15], geometry [30], semantic information [31], material properties [32, 33], human shape and appearance [34], and robotics [35, 36, 37, 38, 39, 40]. Neural Fields have been used in robotics to represent 3D geometry and appearance with applications in grasping [41, 42], trajectory planning [36], object pose estimation and refinement [10, 43, 24, 44], object and surface reconstruction from sparse and noisy data [5, 45], multi-modal perception [46], localization [47] and SLAM [48, 49]. For an overview of recent methods and applications please consult [18].

**Neural Fields for Shape Representation**. These methods represent shapes in the weights of the neural networks, and vary depending on the underlying signal used to encode the 3D space, e.g. occupancy [11, 5], Signed Distance Functions [8, 9, 10], density and radiance [12, 15]. A second distinction comes from the target domain, with some methods overfitting to a single scene/object [12, 20, 21], and other methods learning a generalizable prior over entire categories of shapes, e.g. as a generalizable latent space of Signed Distance Functions [8, 9, 10], as a convolutional prior over grid cells [11, 5] or image pixels [13], as weights of a kernel learned from data [50, 23, 51], as an object centric shape [24] and/or appearance prior [52]. The modeling power of these methods can be further improved by modulating the input coordinates with period functions [53, 54], while rendering speed, training time and networks size can be improved by factorizing the scene tensor into multiple low-rank components [55], by utilizing multiple small size MLPs [56], by training on Sparse Voxel Fields [57] or via multiresolution hash input encoding [21]. A number of methods employ an octree datastructure to guide learning towards occupied areas of space [29, 20, 22]. Closest to our method, NGLOD [20] also uses an octree to adaptively fit shapes to multiple Levels-of-Detail (LoDs), however unlike [20] our method can represent multiple shapes. Additionally, thanks to our recursive scheme, we only need to store root level latents and not the entire grid as is the case for [20]. Recursive parameterizations have also been employed in the context of radiance fields [58] or for 3D shape representation [22]. Unlike [22], we use a lightweight decoder-only recursive architecture to represent high quality shapes as dense oriented point clouds through which we can extract the encoded surface in real-time. Its capacity can be easily extended by simply modifying the size of the input latent vector and it is capable of storing additional attributes (such as material information) at minimal cost.

**Differentiable Rendering** refers to the ability to render an image and back-propagate training signal from the image back to the underlying representation; for an overview of latest methods please refer to [59, 60]. This allows 3D representations of scenes to be learned using only 2D supervision [61], generative models of objects [62], compositionality [63, 64, 65], learning from data in the wild [66, 67, 68] or test-time adaptation [10]. However, in the context of 3D shape representation, extracting the underlying surface from the implicit field typically involves expensive operations such as volume rendering [26], sphere tracing [27] or ray marching [28]. Our method maintains differentiabilty with respect to the input and can thus be optimized given partial 3D data as well as 2D images and additionally we output the underlying surface by design, foregoing the need for expensive operations for surface extraction.

## 3 Methodology

**Preliminaries**. Our approach takes as input a set of shapes $S = \{S_1, \ldots, S_K\}$ and learns a space of implicit surfaces that represents the input. Each shape $S_K$ is represented by an oriented point cloud consisting of points $\{\boldsymbol{p}_i^{GT} \in \mathbb{R}^3\}_{i=1}^{|S_K|}$ and associated normals $\{\boldsymbol{n}_i^{GT} \in \mathbb{R}^3\}_{i=1}^{|S_K|}$, where $|S_K|$ denotes the number of points in $S_K$.

**Formulation**. Our method represents each shape with a $D$-dimensional latent vector denoting the root node of an octree which is traversed recursively, up to a predefined Level-of-Detail (LoD) $M$. We seek to regress three functions to recursively reconstruct 3D geometry, parameterized by neural networks: $\phi : \mathbb{R}^D \to \mathbb{R}^{8D}$ for latent subdivision; $\psi : \mathbb{R}^D \to \mathbb{R}^5$ for mapping the latent space to

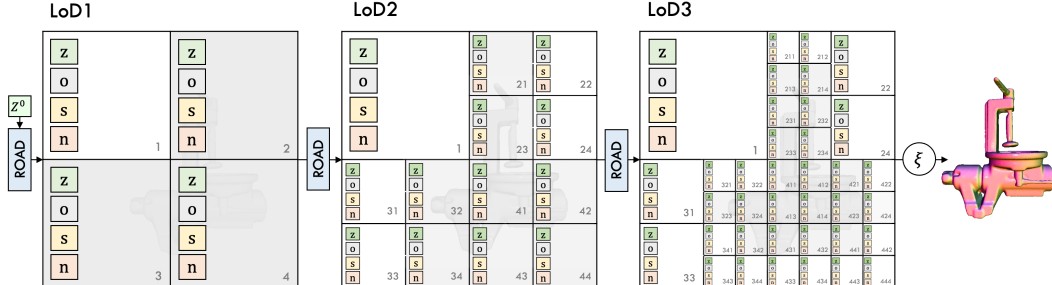

Figure 2: Our method extracts object surfaces by performing octree traversal. Starting from a latent vector of the parent cell ROAD extracts latent vectors for all children cells together with their occupancy, local SDF and surface normal estimates. It efficiently extends to large datasets while retaining high surface reconstruction quality.

surface geometry (occupancy, signed distance to surface, surface normal); and $\xi : \mathbb{R}^7 \to \mathbb{R}^3$ for zero iso-surface projection. The hyperparameter $M$, the dimension of the latent space, is linked to the capacity of our representation as a function of the dataset size. Given a latent vector $z^m \in \mathbb{R}^D$, where $0 \leq m \leq M$ denotes the LoD of the latent vector, we define:

$$\phi(z^m) \to \{z_i^{m+1}\}_{i=1}^8 \qquad (1)$$

as the function that performs a traversal of the latent space given input latent $z^m$ and outputs a latent vector $z_i^{m+1} = \lfloor \phi(z^m) \rfloor_i$ for each of the 8 possible children, where $\lfloor \phi \rfloor_i$ denotes the ith output of $\phi$.

By definition, $\phi$ projects back to the latent space of dimension $D$, i.e. $z_i^{m+1} \in \mathbb{R}^D$, forcing the resulting latent to encode both high and low level information. This formulation allows ROAD to simply pass back the resulting latent to our recursive function, i.e. through $\phi(z^{m+1})$, while propagating coarser, higher-level information along the latent space. A latent $z^0$ at the lowest LoD therefore encodes the geometry of an entire object. Unlike other octree-based methods [20] that store all the octree latents, our formulation allows us to record only the root level latents for each input shape, i.e. $\mathcal{Z}^0 = \{z_k^0\}_{k=0}^K$. We provide ablations for other formulations of $\phi$ in Section 4.

We employ $\psi$ to map any latent vector to underlying surface geometry as follows:

$$\psi(z^m) = \{o^m, s^m, n^m\} \qquad (2)$$

The output of $\psi$ consists of $o^m \in (0,1)$ — the occupancy estimate denoting whether to continue expanding this cell further; $s^m \in (-1,1)$ — the constrained signed distance value from the center of the cell to the surface of the object; and $n^m \in \mathbb{R}^3$ — the surface normal vector. To extract the surface information at a particular LoD $m$ starting from a root latent $z^0$ we perform a tree traversal as follows:

$$\psi(\underbrace{\phi(\ldots(\phi(z^0)))}_{m \text{ times}}) = \{o^m, s^m, n^m\} \qquad (3)$$

Note that for clarity we omitted the notation $\lfloor \cdot \rfloor$ in Eq.3, however after each subdivision the appropriate child latent is selected, according to the desired branch of the octree to be expanded. We highlight that unlike other SDF based methods that implicitly encode surfaces [8], our method does not take as input Euclidean coordinates, and the mapping between different LoD levels is achieved entirely via the learned latents $z^m \to z^{m+1}$ which are forced to encode both local structure as well as global shape information. Moreover, unlike grid-based representations [11, 5] we fully exploit the sparse nature of the underlying octree representation, using the occupancy $o_i$ to recursively expand only occupied cells.

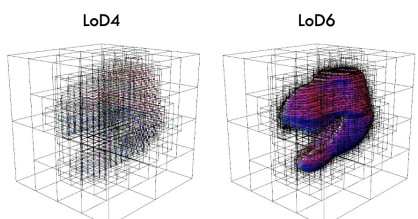

Figure 3: ROAD enables surface points to be directly extracted at different levels of detail.

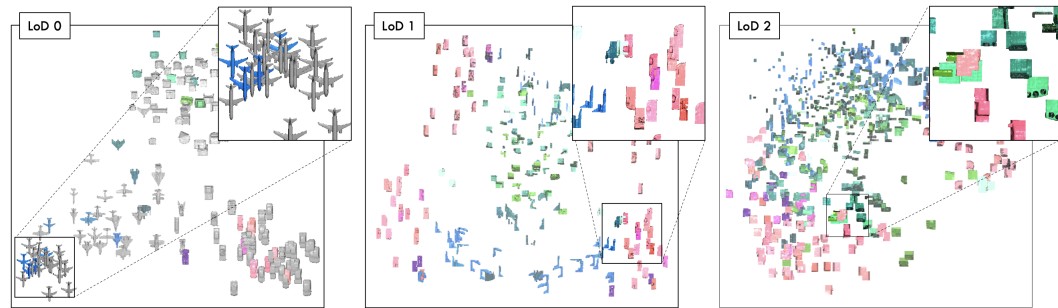

Figure 5: **Latent space structure.** We use principle component analysis to visualize the encoded geometries of the ShapeNet150 dataset in two dimensions. Object color is related to object instance, with objects of the same class having similar colors, and is carried through each LoD. For visualization purposes, grey objects in LoD 0 are not propagated to the higher LoDs. Similar latent vectors encode similar geometries resulting in a clear class separation at LoD 0. Similar areas of the projected latent space become increasingly shared by the different classes at higher LoDs, suggesting that our approach efficiently encodes object geometry by learning common geometric primitives.

**Shape Recovery**. To recover the final shape, we perform the zero-isosurface projection (see Fig. 4), using distance $s_i$, surface normal estimates $\boldsymbol{n}_i$, and voxel cell center coordinates $\boldsymbol{x}_i$ at the desired LoD. The voxel centers are determined and tracked whenever we subdivide a cell. To extract the object surface at LoD $m$, we use the following equation: $\boldsymbol{p}_i^m = \xi(\boldsymbol{x}_i^m, s_i^m, \boldsymbol{n}_i^m) = \boldsymbol{x}_i^m - \alpha^m \boldsymbol{n}_i^m s_i^m$, where $\alpha^m$ is a value that scales $s^m$. In our experiments we use a scaling factor equal to the voxel size at LoD $m$,

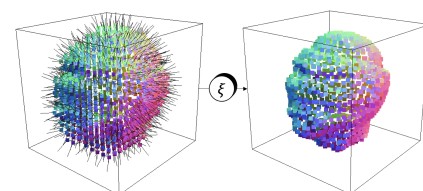

Figure 4: Zero-isosurface projection.

i.e. $\alpha^m = 2/(2^m)$. Once the projection is performed for all the query points at a particular LoD, we get a dense surface point cloud that is differentiable back to the input latent vector $\boldsymbol{z}^0$. This property allows us to perform optimization to complete partial shapes based on the prior encapsulated in the network. An overview of our complete pipeline is provided in Algorithm 1.

**Architecture and Training**. We parameterize the functions $\phi, \psi$ with a single MLP with parameters $\theta$, choosing a SIREN-based [69] network as periodic activation functions have been shown to be more capable at representing fine details. Our MLP uses a single layer encoder and multiple 2-layer decoder heads to output occupancy $o$, SDF $s$, surface normals $\boldsymbol{n}$. All hidden layers are 512-dimensional. We train our method by optimizing both the latent vectors $\boldsymbol{\mathcal{Z}}$ and the parameters $\theta$ of the MLP using the Adam solver with a learning rate of $6 \times 10^{-5}$.

---

**Algorithm 1:** Octree-based recursive surface extraction

**Input:** $M$ maximum recursion depth, $\boldsymbol{\mathcal{Z}}^0 = \{\boldsymbol{z}^0\}$ object latent vector
**Output:** $\mathbf{p} \in \mathbb{R}^3$ surface points,
```
/* Recursively subdivide voxels until desired LoD is reached        */
```
1 **for** $m \in \{1, \ldots, M\}$ **do**
2 $\quad$ $\boldsymbol{\mathcal{Z}}^m \leftarrow \{\}$
3 $\quad$ **for** $\boldsymbol{z}^{m-1}$ *in* $\boldsymbol{\mathcal{Z}}^{m-1}$ **do**
4 $\quad\quad$ $\{\boldsymbol{z}_i^m, o_i^m, s_i^m, \boldsymbol{n}_i^m\}_{i=1}^8 \leftarrow \text{ROAD}(\boldsymbol{z}^{m-1})$ ; $\qquad$ // recursive subdivision
5 $\quad\quad$ $\boldsymbol{\mathcal{Z}}^m \leftarrow \boldsymbol{\mathcal{Z}}^m \cup \{\boldsymbol{z}_i^m | o_i^m \geq \theta\}$ ; $\qquad\qquad$ // add occupied latents
6 $\quad$ **end**
7 **end**
```
/* Extract object shape                                             */
```
8 **for** $\boldsymbol{z}^m$ *in* $\boldsymbol{\mathcal{Z}}^m$ **do**
9 $\quad$ $\mathbf{p} \leftarrow \mathbf{p} \cup \xi(\text{ROAD}(\boldsymbol{z}^m))$
10 **end**
11 **return** $p$

| Method | ShapeNet150 | | | Thingi32 | | |
|---|---|---|---|---|---|---|
| | Storage (MB) (↓) | gIoU (↑) | Chamfer (↓) | Storage (MB) (↓) | gIoU (↑) | Chamfer (↓) |
| DeepSDF [8] | 1052.6 | 86.9 | 0.316 | 224.6 | 96.8 | 0.053 |
| FFN [54] | 301.6 | 88.5 | 0.077 | 64.3 | 97.7 | 0.033 |
| SIREN [69] | 151.3 | 78.4 | 0.381 | 32.3 | 95.1 | 0.077 |
| Neural Implicits [74] | 4.4 | 82.2 | 0.500 | **0.9** | 96.0 | 0.092 |
| NGLOD [20] | 185.4 | 91.7 | 0.062 | 39.6 | **99.4** | 0.027 |
| Ours / LoD6 | | 86.3 | 0.175 | | 96.4 | 0.138 |
| Ours / LoD7 | | 94.2 | 0.067 | | 98.4 | 0.045 |
| Ours / LoD8 | **3.8** | 94.9 | 0.041 | 3.2 | 98.7 | 0.022 |
| Ours / LoD9 | | **94.9** | **0.036** | | 98.7 | **0.017** |

Table 1: **Shape Reconstruction**. This table shows reconstruction and compression comparisons against two datasets. As opposed to the baselines, our method trains a single model for the entire dataset, while still outperforming them in terms of reconstruction quality.

To supervise training, we define losses for each of the decoder levels at every LoD. Occupancy loss $\mathcal{L}_o$ as a binary cross entropy, whereas SDF $\mathcal{L}_s$ and surface normals $\mathcal{L}_n$ losses minimize the $l_2$ distance between respective predictions and ground truth values. The final loss is formulated as:

$$\mathcal{L} = \sum_{m \in M} w_o \mathcal{L}_o^m + w_s^m \mathcal{L}_s^m + w_n \mathcal{L}_n^m, \tag{4}$$

where $w_o = 1$, $w_s^d$ is a function returning an inverse voxel radius for level $m$, and $w_v = 0.1$.

**Curriculum Learning**. When it comes to training on large datasets or on datasets with high resolution models requiring high LoDs, the vanilla training procedure requires much more time to converge as opposed to training on simpler datasets. To alleviate this problem we introduce a curriculum training procedure. Instead of initiating training from the desired final LoD, we change the first LoD to be lower (we use LoD 3 in our experiments) and keep track of the mean of predicted occupancy confidences at a given level $m$. Lower LoDs are faster to train on due to a smaller number of latents to optimize. Moreover, it makes it faster to sample random training points at each iteration, thus further accelerating the training procedure. To compute our confidence score, we first take a softmax over two occupancy values for all estimated voxels at a given LoD, and then take a max value between all the pairs. If our average confidence $\theta$ is high enough in either occupancy or non occupancy, we jump to the next LoD and repeat this procedure until the final LoD is reached. In our experiments, we show that this simple technique helps to accelerate the training process especially when training on large and high resolution datasets. We set $\theta = 0.95$ in our experiments.

## 4 Experiments

To demonstrate the 3D reconstruction and compression capabilities of our approach we run a number of experiments on the ShapeNet [70], Thingi10K [71], Google Scanned Objects [72], and the AccuCities [73] datasets. We report reconstruction metrics as measured by the Chamfer distance (multiplied by $10^3$) as well as interesection over union over points uniformly sampled in the bounding volume of the ground truth shape.

**Reconstruction**. We follow the protocol of [20] and train on a subset of 150 objects from ShapeNet [70] and a subset of 32 objects from Thingi10K [71]. We report results in Table 1. We note that the baselines we compare against train one model for each shape in the dataset (i.e. [20] trains 32 networks for each object in Thingi32). Owing to the efficient recursive decoding scheme implemented by our method, we can train a single model for each dataset and still be competitive in terms of network size and reconstruction accuracy. We set the latent size $D$ to 64 for Thingi32 and to 96 for ShapeNet150 respectively. Indeed, our network achieves state-of-the-art reconstruction results of 0.036 Chamfer distance and an

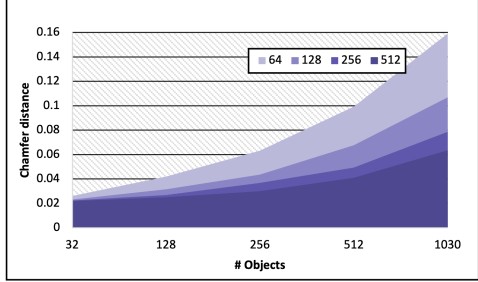

Figure 6: Chamfer distance vs data quantity for different latent vector sizes.

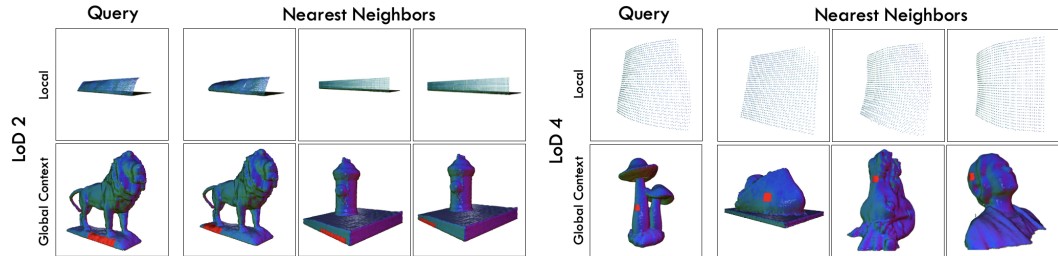

Figure 7: **Nearest neighbors.** We visualize the geometry encoded by example latents, as well as the geometry encoded by the nearest neighbors within the same LoD across all objects in the Thingi32 dataset, determined by the Euclidean distance metric on the latent vector space (top row). We color the points in the octree cell corresponding to the latent, showing that our approach enables latents with similar shapes to be used at different global positions (bottom row).

IoU of more than 94% on ShapeNet150 with a network of size 3.8MB: this amounts to a compression of more than 99%, as the original mesh dataset measures 630MB. We report a similar compression ratio on Thingi32 (473MB) while achieving state-of-the-art reconstruction accuracy of 0.017 Chamfer distance and a competitive IoU of 98.7%.

We further explore the reconstruction capabilities of our method by encoding a model from the AccuCities [73] dataset: a neighborhood from London consisting of 1.9 million triangles and requiring 252MB of disk space. We set the latent size $D$ to 512. Qualitative results are shown in Fig.1 with additional images in the supplementary. Quantitatively we achieve a Chamfer distance of 0.04 when comparing against the ground truth model, while using a network of size 11 MB.

**Scaling to larger datasets**. For this experiment we use the entire Google Scanned Objects [72] dataset consisting of a total of 1030 object models. We introduce training splits of different sizes (32, 128, 256, 512, 1030) to study the relation between the dataset complexity and the latent vector size, a novel hyperparameter specific to our formulation. We restrict our analysis into the scaling properties of our network to the dimension of the latent space, and mention that other methods that are generally applicable to machine learning models (i.e. number of layers, training schedule, etc.) can be used to further tweak the performance of our method. Our results are recorded in Fig. 6: for each split we train networks with increasing latent vector sizes (64, 128, 256, 512) and record the resulting Chamfer distance by comparing the reconstructed models with the ground truth models of that specific split. We note a strong correlation between dataset complexity and latent size. Specifically, our results indicate that our method can efficiently scale to an increasing number of shapes by only modifying the latent vector size while keeping the network parameterization intact.

**Latent Space Analysis**. To qualitatively analyze the properties of the learned latent space, we project the latent space at specific LoDs into two dimensions via principle component analysis to visualize the encoded geometries of the ShapeNet dataset, as shown in Fig. 5.

We observe that objects of the same class are spatially close in the projected space at the top level LoD, demonstrating that similar latent vectors encode similar geometries. Furthermore, at higher LoDs, similar areas of the projected latent space are increasingly shared by the different classes, suggesting that our approach efficiently encodes object geometry by learning geometric primitives common in the dataset.

We also visualize the nearest neighbors of specific latent vectors from the network trained on Thingi32 to LoD 9 in Fig. 7. We show that similar latent vectors can represent the local geometry at different object coordinates, without explicit positional encoding. Increasing the LoD also intuitively reduces the geometric complexity represented by the latent, as seen by the 3D edge feature from LoD 2 and the oriented patch feature from LoD 4.

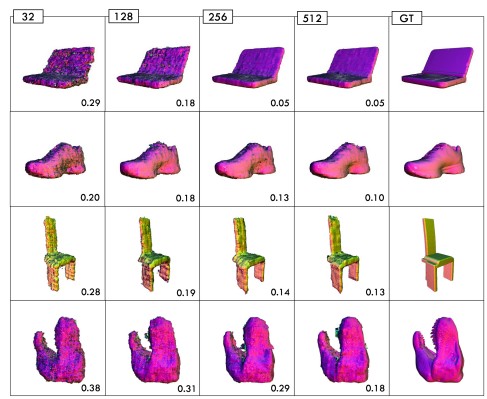

Figure 8: **Generalization.** We demonstrate our method's increasing generalization ability when trained on many objects.

| Superposition | Storage (MB) | gIoU | Chamfer |
|---|---|---|---|
| Direct | 3.2 | 98.7 | 0.017 |
| Addition | 3.2 | 96.9 | 0.039 |
| Concatenation | 17.9 | 99.4 | 0.013 |

Table 2: **Latent Fusion.** Here we compare different ways to propagate latent vectors to the next LoDs.

| Surface density | Low | Medium | High |
|---|---|---|---|
| Sphere tracing | 5 min | 6 min | 10 min |
| Marching cubes | 0.1 s | 0.9 s | 6 s |
| Ours | **11 ms** | **13 ms** | **17 ms** |

Table 3: **Inference time.** Our method extracts object surfaces in real time, significantly outperforming the state of the art.

**Generalization**. In this experiment we demonstrate the generalization capabilities of our method. We take four networks from the data compression experiment each trained on a different split of the Google Scanned Objects dataset (32, 128, 256, and 512 objects) and optimize latent vectors to fit unseen models from the same as well as other datasets (Thingi32, ShapeNet150) while keeping network weights frozen. The results are shown in Fig. 8. Reconstruction quality plotted in terms of Chamfer distance shows increasing generalization capability for networks trained on more models.

**Surface Extraction**. By design, our method differentiably extracts object surfaces in real time with minimal memory overheads. In comparison, NGLOD [20] or other SDF-based [8] methods require either an expensive sphere ray tracing or non-differentiable marching cubes to extract surface. Table 3 compares inference times when extracting the object surface at different levels of density. For our method we perform inference up to LoDs 6, 7 and 8 respectively, which corresponds to approximately 20000, 80000 and 300000 surface points, respectively for the Thingi32 models. We compare against a marching cubes baseline from [8] and a sphere tracing baseline from [20], and we iterate until the desired number of surface points is sampled. For a fair comparison with our method we extract 20000, 80000 and 300000 using both sphere tracing and marching cubes. We note that our method extracts the object surface up to 3 orders of magnitude faster that the sphere tracing baseline; our method requires *a total of 1-2 seconds* to extract object surfaces for the *entire* ShapeNet150 dataset. The marching cubes baseline, while faster than sphere tracing, is still not real-time capable and not differentiable. Finally, we observe that the results for this experiment were obtained on a single A6000 GPU, without any optimization.

**Latent Vector Fusion**. As described in Section 3 and in Fig. 2, our method propagates information through the latent space via the recursive function $\phi$ (see Eq.1). Here we explore different forms of latent subdivision: addition and concatenation, with the complete definitions provided in the supplementary. While addition does not change the dimension of the latent vector $D$, it explicitly defines how information is propagated from parent to child latent (i.e. via addition). Conversely, concatenation makes $D$ increase with each recursion level, and information is directly copied as we traverse the latent space. This introduces significant modifications to the underlying neural network architecture, requiring specialized networks at each LoD. The results of this ablative analysis are summarized in Table 2. As expected, contcatenation serves as an upper bound for performance and it achieves the highest performance but requires $5\times$ more storage space, and a more complicated formulation. Although similar in formuation to direct regression, the addition version of our method achieves poor results, which we attribute to the artificial constraint imposed on how information is propagated through the latent space.

## 5   Discussion

**Limitations and Future Work**. Our representation currently only supports 3D geometry. For future work, we would like to explore its extension to other modalities (object color and material properties) as well as representations (images and radiance fields [12]). Another interesting direction that could be explored is a combination of our pipeline with different downstream tasks (object detection and pose estimation). Our representaton is fully differentiable and thus allows the propagation of useful 3D gradients for shape optimization given partial information.

**Conclusion**. We presented a novel recursive implicit representation to effectively represent and compress 3D geometry by framing it as the traversal of an implicit octree in a learned latent space. It extracts geometry in real-time and scales to large datasets while retaining high reconstruction quality. As a result, we outperform state-of-the-art reconstruction results on the ShapeNet150 and Thingi32 datasets, even when compared to methods training a single network per model. Our analysis of the representation explores the structure of the latent space and presents a scaling law defining a relationship between latent space dimension, dataset size and reconstruction accuracy.

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
