# OpenReview forum: "ROAD: Learning an Implicit Recursive Octree Auto-Decoder to Efficiently Encode 3D Shapes"
_robot-learning.org/CoRL/2022/Conference — CoRL 2022 Poster_

### Official Review · Reviewer_SYPp · 2022-07-22

**Originality:** Good
**Technical Quality:** Very Good
**Clarity Of Presentation:** Very Good
**Impact:** 3

**Recommendation:**

Strong Accept: I recommend accepting the paper and will argue for my recommendation even if other reviewers hold a different opinion.

**Summary:**

The paper proposes an octree-based implicit representation of 3D shapes. Explicit representations of such data (point clouds, meshes) present a high memory footprint that increases with resolution, while the proposed approach represents a shape with a single latent vector and a decoding neural network able to generalize to multiple shapes, resulting in a very efficient compression ratio while preserving a high accuracy.

**Issues:**

As explained above, I believe that details are lacking concerning the input data: oriented 3D point cloud, meshes ? In particular, more explanation about how SDF labels are computed from a sparse point cloud would make the reproducibility easier.

I am quite confused about the "curriculum learning" description. L178 "Instead of initiating from the desired final LoD, we change the first LoD to be lower". Does it mean that only the LoD at lower resolutions are trained in the first place, and then finer details are introduced ? L182 "Once we reach a desired probability threshold theta, ..." This notation is used for network weights earlier in the paper and it is not clear to me which probability is mentioned. I think the writing of this section can be improved.

I am curious about answers and comments about my remarks in the Questions/Suggestions section, however I am not asking for additional experiments concerning them during the rebuttal.

**Quality Of The Limitations Section:**

Limitations are addressed clearly

**Reviewer Expertise:**

3: The reviewer is fairly confident that the evaluation is correct

**Robotics Focus:**

Highly relevant to robotics but no hardware experiments

**Strengths And Weaknesses:**

**Strengths:**
* Quantitative results show a clear improvement over prior work, considering the trade off between reconstruction accuracy, storage requirements and inference time. The high compression ratio, coupled with a fast decoding time make the contribution a very appealing solution for embedded robotics downstream tasks.
* The proposed method does not simply overfit a neural network on each 3D shape but can be trained with multiple complex shapes simultaneously. When the training dataset is sufficiently diverse, the approach exhibits good generalization properties to unseen shapes, showing that the network does not only memorize training shapes but learns to combine geometric primitives at different level of details. This claim is supported by several convincing experiments.
* The recursive decoding scheme based on an octree data structure is novel and relevant: it enables to limit the number of network parameters because decoding functions are shared between different resolutions. At inference time, the user can choose the desired level of detail, depending on throughput and accuracy requirements.

**Weaknesses:**
* Details are missing concerning how ground truth labels (occupancy, signed distance function and normal vectors of each cell) are defined. The paper states that input shapes are represented as an oriented point cloud (L112), but datasets used in experiments seems to provide meshes (L204). For the sake of reproducibility, I think that the data format of the input dataset should be clarified. While the preprocessing from point cloud to surface might be done with classical approaches, more details would be useful for an uninformed reader.
* No experiments are conducted on downstream robotics tasks (stated in the limitations section). The differentiability of the resulting surface w.r.t the input latent vector is an interesting property which might enable to learn the shapes from weaker (2D) supervision signals.  This is suggested in the related work section (L106), but not supported by empirical experiments.

**Questions/Suggestions:**
* As a conclusion of the proposed experiments, I understand that the decoding network embeds a mapping from latent keys to the geometrical primitives present in the training set, whereas the input vector encodes these latent keys for different resolutions in a compact way. As a result, a new shape can be integrated in the framework without modifying the network weights ("generalization" experiment), as long as all geometric primitives present in the shape have been observed in other shapes during training. Is that correct ?
* One drawback of implicit representations is the offline training time. How long training takes to fit a single shape ? A large dataset ? How fast is it to optimize only the input vector of a new shape in a trained network, compared with retraining everything ?
* I would expect that simple augmentations during training (random 3D rotations and crops of the shapes) should help to improve the generalization ability of the decoding network.
* Do you think that the proposed method could be applied on 2D signals using a quadtree data structure ?

**Summary Of Recommendation:**

I recommend "Strong Accept" for this paper. I believe that the contribution is quite strong, with a clear writing and thorough experiments. I believe that the paper is suitable for publication in its current form, even if some minor details should be added for reproducibility.

---

> ### Author Response · Authors · 2022-08-24
> **Response to Reviewer SYPp**
>
> **Comment:**
>
> We would like to thank reviewer SYPp for the feedback and useful suggestions. We also appreciate the acknowledgment of our work’s novelty, strong quantitative results, relevance to robotics applications, and good generalization properties. Below we answer the questions raised in the feedback.
>
> > Ground truth labels
>
> In our experiments, we extract ground truth labels from meshes and pointclouds, and generally require dense surface points to obtain accurate labels. In practice, the occupancy label of a voxel at a particular LoD is determined by querying whether a point exists within the voxel of interest, and the SDF value and normal are extracted from the nearest neighbor to the voxel center. We observe that these same quantities could also be extracted from an object represented by an SDF. Additionally, we pre-compute and store these annotations once per dataset over all LoDs. We will update the paper to include these details.
>
> > Downstream robotics tasks
>
> As mentioned in our paper (Limitations section), and as we show through our generalization experiment, our method is fully differentiable allowing us to reconstruct novel shapes from point cloud data. For the rebuttal we have further expanded this experiment showing good performance on sparse and noisy data as well, as shown in A3 of the main answer. We believe this is a key capability of a robotic perception pipeline. In future work, we aim to further integrate our method with other relevant downstream applications, which we cover in more detail in A1 of our general response.
>
> > Reusability of learned geometric primitives
>
> As demonstrated in our generalization experiment, ROAD is capable of generalizing to completely out-of-domain shapes. We have further extended this experiment for the rebuttal, please refer to A3 in the main answer. Our method’s generalization capability is made possible by efficiently reusing geometric primitives from different LoDs learned during training. This is also demonstrated in the Latent Space Analysis section of the main paper and the supplementary video, where similar areas of the projected latent space are increasingly shared by the different classes.
> We also attach a video where we visualize PCA projected features of a ROAD model trained on a sequence of three objects being translated inside a unit cube. We observe that these features remain largely stable reaffirming primitive reusability and translation invariance properties of our approach.
>
> > Training time comparison
>
> Please refer to A2 of our general response.
>
> > I would expect that simple augmentations during training (random 3D rotations and crops of the shapes) should help to improve the generalization ability of the decoding network.
>
> We believe that introducing augmentations would indeed increase generalization performance since our network is not rotationally invariant. In the attached animation we demonstrate what PCA projected features look like when rotating the object. Alternatively, one could potentially combine our approach with rotationally equivariant architectures (e.g., [Vector Neurons](https://cs.stanford.edu/~congyue/vnn/)). We leave this to future work.
>
> > Do you think that the proposed method could be applied on 2D signals using a quadtree data structure?
>
> Yes, the recursive idea of our proposed method is not limited to the 3D domain and could be applied to 1D, 2D, and potentially to higher dimensional data as well. However, given that our method takes advantage of data sparsity (i.e., only the surface is encoded), it would be best suited for sketches rather than dense images.
>
> > Curriculum learning
>
> In the curriculum learning, we start by training iROAD at a lower LoD and only change to the next level when the mean of predicted occupancy confidences at the current LoD is larger than a threshold. In particular, to compute our probability score, we first take a softmax over two values for all estimated voxels at a given LoD, and then take a max value between all the pairs. If our average confidence is high enough in either occupancy or non occupancy, we switch to the next level. We repeat this procedure every N epochs until a desired resolution is reached. This allows us to accelerate the training process and concentrate on essential shape features during initial training stages. We will include a detailed description of this procedure in the revised version of the paper.
>
>
> **Zip File:**
>
> /attachment/0302dafd95af5c335544aa594a5bb46b2ec78b61.zip

---

> > ### Comment · Reviewer_SYPp · 2022-08-25
> > **Feedback**
> >
> > Thanks for the answers. I believe my concerns have been addressed and I will keep my 'Strong Accept' recommendation.
> >
> > As a note, I think the following papers are also relevant as robotics applications of neural fields:
> >
> > * iMAP: Implicit Mapping and Positioning in Real-Time, ICCV 2021
> > * NEAT: Neural Attention Fields for End-to-End Autonomous Driving, ICCV 2021
> > * LENS: Localization enhanced by NeRF synthesis, CoRL 2021

---

> > > ### Author Response · Authors · 2022-08-25
> > > **Re: Feedback**
> > >
> > > Thank you very much for your response and for providing additional relevant works! We will update the paper to include these references into related work.

---

### Official Review · Reviewer_XRB2 · 2022-07-31

**Originality:** Very Good
**Technical Quality:** Very Good
**Clarity Of Presentation:** Very Good
**Impact:** 2

**Recommendation:**

Weak Accept: I recommend accepting the paper, but will not argue for my recommendation if the majority of other reviewers have a different opinion.

**Summary:**

The paper proposes a hierarchical, recursive latent space to learn an object shape representation.
Starting from a root latent vector, a neural network recursively maps this latent vector into 8 more latent vectors refining the initial one. This way, the authors achieve higher surface reconstruction capabilities while keeping computation and memory requirements low.

**Issues:**

Please clarify why it is sufficient to have a single root latent vector to represent fine details and why this is not possible with a monolithic model.

**Quality Of The Limitations Section:**

Additional details required

**Reviewer Expertise:**

4: The reviewer is confident but not absolutely certain that the evaluation is correct

**Robotics Focus:**

Irrelevant to robotics

**Strengths And Weaknesses:**

Strengths:
The paper is well written and I enjoyed reading it.
The idea, to my knowledge, of having a hierarchical latent space for representing an object shape is novel.

Weaknesses:
However, I have two concerns:
1) The biggest concern about this paper is the relation to robotics. I have no idea why this was submitted to CoRL.
2) The root latent vector has to contain all information about the object. Does this not counteract the idea of having an hierarchical representation of an object?

**Summary Of Recommendation:**

Unfortunately, there is no relation to robotics in this paper. I have chosen weak accept, as I generally think the paper deserves publication, but I highly doubt that CoRL is the right venue for this work.

---

> ### Author Response · Authors · 2022-08-24
> **Response to Reviewer XRB2**
>
> We would like to thank reviewer XRB2 for the feedback and useful suggestions. We also appreciate the acknowledgment of the novelty of our hierarchical latent space for modeling shapes and that our paper is well written. Below we answer the questions raised in the feedback.
>
> > Relation to Robotics
>
> Please refer to A1 of our general response.
>
> > The root latent vector has to contain all information about the object. Does this not counteract the idea of having an hierarchical representation of an object?
>
> Neural fields methods, including our baselines, map xyz coordinates and optionally a global latent vector, to some property of the scene (i.e. occupancy, signed distance). iROAD does not take as input xyz coordinates and instead learns to recursively extract object geometry by traversing an implicit octree in latent space. This formulation allows our method to efficiently reuse geometric primitives common in the dataset as also shown in the Latent Space Analysis section of the main paper, demonstrating that similar areas of the projected latent space are increasingly shared by the different classes. Our root latent vector therefore acts as a guide in shape extraction by reusing common parts, rather than explicitly compressing a full object representation. This property enables high compression rates, ability to accurately recover increasingly large datasets, and a real-time shape extraction.

---

> > ### Comment · Reviewer_XRB2 · 2022-08-28
> > **thank you for your reply**
> >
> > > Our root latent vector therefore acts as a guide in shape extraction by reusing common parts, rather than explicitly compressing a full object representation.
> >
> > This is highly confusing. Under the assumption that your approach can be scaled to represent any arbitrary shape, then the root latent vector must contain, in my understanding, the full object representation.
> > Can you please clarify that?

---

> > > ### Author Response · Authors · 2022-08-28
> > > **Response to Reviewer XRB2**
> > >
> > > Thank you for the question. To clarify: yes, the root latent contains all the information needed to represent an entire object. Additionally, our answer highlights the distinction between our method and methods like DeepSDF, which use a single latent per shape: these methods encode the shape and the space around it continuously, and they are evaluated multiple times at spatial coordinates to extract the underlying quantity encoded (i.e., in the case of DeepSDF that is the signed distance value to the surface). Our method is decoupled from coordinates, and instead learns to recursively extract object geometry by traversing an implicit octree in the latent space. Our qualitative results (Fig. 5 - Latent space structure, main paper) suggest that our method learns to reuse geometric primitives, and this is particularly noticeable at higher LoD levels which encode fine details that are shared across shapes. These results indicate that our method learns to encode geometry in a latent space that encodes geometric primitives at different levels effectively, and allows their reuse across shapes, which leads to high-quality reconstruction and efficient compression (Table 1, main paper).
> > >
> > > Please let us know if this answers your question or if you have any remaining questions - we'd be happy to answer them and continue discussing.

---

### Official Review · Reviewer_NizH · 2022-08-01

**Originality:** Very Good
**Technical Quality:** Very Good
**Clarity Of Presentation:** Very Good
**Impact:** 4

**Recommendation:**

Strong Accept: I recommend accepting the paper and will argue for my recommendation even if other reviewers hold a different opinion.

**Summary:**

This paper presents a highly-efficient data structure for encoding multiple 3D shapes in a low dimensional latent space. The key idea is to use a hybrid neural network / octree structure to store signed distance function (SDF) values. The latent vector defines both the Octree structure and, via a small MLP, the SDF values. The results show that the method exhibits excellent compression ratios and reconstruction accuracy.

**Issues:**

Please address the questions in weaknesses above.

**Quality Of The Limitations Section:**

Limitations are addressed clearly

**Reviewer Expertise:**

4: The reviewer is confident but not absolutely certain that the evaluation is correct

**Robotics Focus:**

Relevant but unlikely to deploy to hardware in near future

**Strengths And Weaknesses:**

Strengths
=========
- ”Neural fields” models are known to be quite limited in terms of the scale of scenes they can represent. Leveraging Octrees therefore has great potential to get neural fields models to scale to large scenes, but a significant challenge lies in how to get the Octree structure to generalize across shapes. This paper presents a solution to this problem by encoding the Octree shape with the latent code.
- The architecture of the model is well designed and elegant. I especially like how the latent code recursively defines the Octree structure via splitting, much like you would construct a traditional octree.
- The paper is very clearly written and nicely illustrated. A lot of effort has been put into communicating and explaining the idea.

Weaknesses
==========
- There is not much discussion in the paper of the tradeoffs or weaknesses of the approach. How long does recursive decoding of the Octree take, for example?
- The genrealization experiments are quite limited. How many sample points of the target object are used during the optimization and how is this affected by sampling fewer points? Can the latent vector be predicted from an image?


**Summary Of Recommendation:**

Overall, this is a good paper with a strong contribution in terms of a new hybrid neural+Octree data structure that can have applications in many areas in robotics and computer vision.

---

> ### Author Response · Authors · 2022-08-24
> **Response to Reviewer NizH**
>
> **Comment:**
>
> We would like to thank reviewer NizH for the feedback and useful suggestions. We also appreciate the acknowledgment of our work’s scalability, generalization across shapes, that our architecture is well designed and elegant and that our paper is very clearly written and nicely illustrated. Below we answer the questions raised in the feedback.
>
> > Weaknesses, Octree recursive decoding time
>
> Apart from the weaknesses mentioned in the Limitations section of the main paper, we would like to highlight that our method is not rotationally invariant. This means that a single local geometry primitive under two different rotations is considered as two separate primitives by our network, and therefore requires learning new primitives for every new rotation. In the attached animation we demonstrate what PCA projected features look like when rotating the object, showing that features are indeed not rotationally invariant. As a potential solution one could combine our approach with rotationally equivariant architectures (e.g., [Vector Neurons](https://cs.stanford.edu/~congyue/vnn/)). We leave this to future work.
>
> In Table 3 of the main paper, we compare different geometry extraction methods and also show decoding times of our method corresponding to LoDs 6, 7, and 8. As can be seen from it, our method allows extracting surfaces in real time, where other methods require significantly more time.
>
> > Extended generalization experiment
>
> Please refer to A3 of our general response.
>
> > Can the latent vector be predicted from an image?
>
> Similar to a number of works using an implicit shape decoder, e.g. [OccupancyNetworks](https://avg.is.mpg.de/publications/occupancy-networks), a latent vector in iROAD can likely be predicted from an image by additionally training an image encoder or, alternatively, combining iROAD with a differentiable renderer (as in [DIST](https://arxiv.org/abs/1911.13225) or [SDFlabel](https://arxiv.org/abs/1911.11288)) and regressing the best fitting latent vector via optimization. We believe this is a promising direction for future work.
>
>
> **Zip File:**
>
> /attachment/eb3e4078c2adcf2d2ca42ca55b57bb505d03abe4.zip

---

### Official Review · Reviewer_dXT1 · 2022-08-02

**Originality:** Good
**Technical Quality:** Very Good
**Clarity Of Presentation:** Very Good
**Impact:** 4

**Recommendation:**

Weak Accept: I recommend accepting the paper, but will not argue for my recommendation if the majority of other reviewers have a different opinion.

**Summary:**

This paper proposes an efficient implicit representation to encode large scale datasets in real time: implicit recursive octree auto-decoder iROAD. The idea is directly learning latent code for each object in a hierarchy manner. During the inference, given the initial latent code $\mathbf{z}_0$, we can call iROAD recursively to get latent code for different level of details (LoD). To speed up training, this paper introduces a curriculum procedure to first train low LoD and gradually jump to higher LoD until the desired/final LoD reached. Experiments on synthetic datasets (ShapeNet and Thingi32) show that the proposed method outperforms existing approaches (e.g., DeepSDF, NGLOD) with a small memory footprint.

**Issues:**

See weakness. Primarly more comparison with baselines on the training time.

**Quality Of The Limitations Section:**

Limitations are addressed clearly

**Reviewer Expertise:**

3: The reviewer is fairly confident that the evaluation is correct

**Robotics Focus:**

Relevant but unlikely to deploy to hardware in near future

**Strengths And Weaknesses:**

**Strengths:**

[S1] The paper is well motivated - compact and accurate representation of 3D shapes are always critial for robotics tasks. However, only a few approaches are proposed (NGLOD) to handle this problem. The idea of iROAD is simple yet very effective -> better than more complicated NGLOD.

[S2] The performance is great - iROAD outperforms to methods training a single network per model significantly. This inspires us that jointly training will encode more data priors and achieve superior performance. I like the generalization experiment which shows that the latent space is able to generalize on unseen objects (seen class).

**Weakness:**

[W1] Training time comparison for iROAD and other baselines (DeepSDF, NGLOD) for encoding the whole dataset. I am concerned that training auto-decoder in hierarchy way would significantly increase the computation load. More concise comparison with baselines especially NGLOD would be highly appreciated.

[W2] Quanlitative results?  It would be great to have more visual comparisons with existing approaches and see the difference in recovered surface (vs DeepSDF or NGLOD).  That would make the results more convincing.

[W3] More experiments on the robustness of iROAD would be more interesting. Currently, all results are conducted on synthetic datasets with dense oriented points. In practice, the points might be sparse and contain noise, it would be very interesting to see whether the joint learning can help overcome the noise and sparisty issue with priors from other objects.

**Others:**
- why not reporting the storage for other LoDs in Table 1. Could we show the results of NGLOD with different LoD level?
- In Table 3, it would be great to be more explicit - showing the performance of NGLOD and iROAD for each level (low, medium and high) and the inference time.
- Good to have: Training time vs number  of objects
- Include more implementation details for baselines

**Summary Of Recommendation:**

The overall idea for this paper is simple but the performance is great. It outperform existing SOTA approach NGLOD and also achieves a large margin than some methods training a single network per model. The inference can be conducted in real time and the model storage is very efficient.

---

> ### Author Response · Authors · 2022-08-24
> **Response to Reviewer dXT1**
>
> **Comment:**
>
> We would like to thank reviewer dXT1 for the feedback and useful suggestions. We also appreciate the acknowledgment of our work’s simplicity and effectiveness, relevance to robotic applications, great performance and generalization capabilities. Below we answer the questions raised in the feedback.
>
> > Training time comparison for iROAD and other baselines
>
> Please refer to A2 of our general response.
>
> > Qualitative results
>
> The attached figure compares surface quality between DeepSDF, NGLOD, and our method. As also backed up by the metrics (cf. Table 1 in the main paper), reconstruction quality of DeepSDF falls behind other methods. NGLOD, on the other hand, demonstrates a very similar visual reconstruction quality to our method, while our method is trained on multiple objects (32 in this visualization). Moreover, DeepSDF and NGLOD use a different geometry extraction algorithm based on expensive and time consuming sphere ray tracing (cf. Table 3 in the main paper). iROAD, on the other hand, directly outputs oriented point clouds by design and as a result extracts surfaces in real time.
>
> > Generalization robustness
>
> Please refer to A3 of our general response.
>
> > NGLOD results for other LoDs in Table 1
>
> We only reported the official best performing NGLOD (LoD5) results in Table 1. At lower levels of detail NGLOD loses in reconstruction quality, but indeed gains in storage efficiency. In particular, given the lowest level of detail - LoD1, NGLOD achieves only 0.343 Chamfer (vs 0.062 at LoD5) for ShapeNet150 dataset, while only needing 96 KB (vs 1356 KB at LoD5) per model, resulting in 14 MB for ShapeNet150 if using different a separate decoder per model. All NGLOD numbers relating to performance and storage are reported directly from the published paper.
>
> > Table 3 extended results
>
> We would like to note that Table 3 compares extraction times of different surface extraction methods. The number of used points per method roughly corresponds to the number of points our method extracts on LoDs 6, 7, 8 — 20000, 80000, and 300000 points. For the sphere tracing baseline, we use NGLOD’s implementation of the surface extraction and use an NGLOD LoD 5 model to sample a predefined number of points. We also note that iROAD LoDs do not correspond to NGLOD LoDs due to architectural differences, making a possible performance comparison within this table ambiguous.
>
> Besides, we extend our comparison by adding a baseline used in the original DeepSDF implementation — marching cubes. While faster than sphere tracing, marching cubes are still not real-time capable and not differentiable. In particular, we report 0.1s for 20000 points, 0.9s for 80000 points, and 6s for 300000 points.
>
> > Implementation details for baselines
>
> We use the original NGLOD implementation as well as NGLOD’s re-implementations of presented baselines: DeepSDF, FFN, SIREN, and Neural Implicits. We will update the paper to include these details.
>
> **Zip File:**
>
> /attachment/54402c5c9f5fb6fbdf2153242e740aea249cd079.zip

---

> > ### Comment · Reviewer_dXT1 · 2022-08-26
> > **Thanks for the additional experiments and detailed response!**
> >
> > I would like to thank the prompt and detailed response. Most of my concerns have been addressed well. I also read the other reviews and the author's response carefully.
> >
> > Considering all reviews and authors' response. I think this paper is a good contribution to the CoRL community and would like to keep my original score.

---

### Author Response · Authors · 2022-08-24
**General Response to AC and Reviewers**

We thank the reviewers and the AC for their positive comments and thoughtful feedback.
We appreciate that they find our approach to be novel (XRB2, SYPp), well-motivated (dXT1), effective (dXT1), well designed and elegant (NizH), and the paper to be clearly written and nicely illustrated (NizH). We also thank their acknowledgement of our strong empirical results and recognition of our method as an appealing solution for embedded robotics downstream tasks (NizH, SYPp).

Below, we respond to the most important questions raised by multiple reviewers, and we provide individual responses to each reviewer addressing any remaining questions. Please let us know if any other questions arise - we are happy to discuss further and answer any remaining points.

---

> ### Author Response · Authors · 2022-08-24
> **Q3. [dXT1, NizH] Generalization robustness**
>
> **Comment:**
>
> **A3.**
>
> |  Object | 0.1% (L3) | 0.3% (L4) | 1.5% (L5) | 6% (L6) | 25% (L7) | 100% (L8) |
> |:-------:|:---------:|:---------:|:---------:|:-------:|:--------:|:---------:|
> | Console |   0.422   |   0.144   |   0.083   |  0.065  |   0.055  |   0.053   |
> | Sneaker |   0.980   |   0.287   |   0.163   |  0.124  |   0.109  |   0.105   |
> |  Chair  |   1.120   |   0.354   |   0.224   |  0.161  |   0.136  |   0.135   |
> |  T-Rex  |   1.361   |   0.467   |   0.248   |  0.224  |   0.189  |   0.180   |
> |   Mean  |   0.971   |   0.313   |   0.179   |  0.143  |   0.122  |   0.118   |
>
> As suggested, we extended the generalization experiment to include multiple grades of sparsity and noise. We use a network trained on 512 dense objects of the Google Scanned Objects dataset. We then optimize latent vectors to fit unseen objects from the generalization experiment. Given a ground truth dense unseen object point cloud we apply sparse supervision computed from the dense GT point cloud to estimate the surface geometry. We optimize the pre-trained iROAD to a lower LoD, i.e., provide a coarser supervisory signal than the network was trained to. We then extract the surface to the highest LoD. LoDs 3 through 7 represent approximately 0.1%, 0.3%, 0.15%, 6%, and 25% of the supervision at LoD8. We observe that even in the case of optimizing only to LoD3, our method is still able to converge to reasonable shapes (see attached Figure "sparsity_ablation.png").
>
> |  Object | Small (L7) | Medium (L6) | Large (L5) | Severe (L4) |
> |:-------:|:----------:|:-----------:|:----------:|:-----------:|
> | Console |    0.053   |    0.058    |    0.096   |    0.567    |
> | Sneaker |    0.106   |    0.114    |    0.164   |    0.657    |
> |  Chair  |    0.139   |    0.172    |    0.232   |    0.714    |
> |  T-Rex  |    0.183   |    0.210    |    0.287   |    0.795    |
> |   Mean  |    0.120   |    0.138    |    0.195   |    0.683    |
>
> We additionally demonstrate how noise affects the generalization performance. Similarly to the  sparsity experiments above, we optimize the pre-trained iROAD network to a lower LoD and additionally we randomly perturb SDF annotations at the final LoD of interest with a uniform noise distribution scaled by the voxel size. This procedure corresponds to adding different levels of metric noise at LoD 7 (small), 6 (medium), 5 (large), and 4 (severe). We then use a network trained on unperturbed data (from the same set of 512 Google Scanned Objects as in the experiment above) to fit to the occupancy and noisy surface annotations at a particular LoD; for all LoDs below the query LoD we supervise only on occupancy. Finally, we visualize the fully extracted object (i.e., at LoD 8). Once again, we observe that our method is robust to introduced perturbations and is able to faithfully reconstruct objects even when noise is introduced (see attached Figure "noise_ablation.png").
>
> **Zip File:**
>
> /attachment/a332aa8e270eadc226c3246e94d36589f1cdaf5b.zip

---

> ### Author Response · Authors · 2022-08-24
> **Q2. [dXT1, SYPp] Training time comparison for iROAD and other baselines**
>
> **A2.**  We use a single A100 GPU for training. Our method's time to convergence depends on the number of objects to be encoded: going from approximately 40 min for a single model, 8 hours for 32 models to around 20 hours for 1030 models of the Google Scanned Objects dataset. However, we did not experience a significant difference between models trained on 256, 512, and 1030 objects, all requiring around 20 hours to converge. This suggests that our method can leverage an increasing number of reusable local components from higher LoDs and trains efficiently as the number of objects increases. NGLOD, on the other hand, doesn’t support training on multiple objects and similarly needs approximately 40 minutes (LoD 5) per model to converge on a single GPU. If we assume a sequential training setup, this would result in 28.6 days of training to converge for 1030 objects.
>
> Finally, if we consider only optimizing the input latent vector using a pretrained network (as in our Generalization experiment) it only takes around 7 min to converge to a single model.

---

> ### Author Response · Authors · 2022-08-24
> **Q1. [AC, XRB2] Relation to Robotics**
>
> **A1.** Neural Fields have been used in robotics to represent 3D geometry and appearance with applications in grasping [A,B], trajectory planning [C], object pose estimation and refinement [D,E], object and surface reconstruction from sparse and noisy data [F,G], and multi-modal perception [H]. We will update our related work section to explicitly highlight neural fields and their applications in robotics.
>
> Further, the reviewers mention our method's relevance to robotics in the following ways: "compact and accurate representation of 3D shapes are always critical for robotics tasks" (dXT1), "a new hybrid neural+Octree data structure that can have applications in many areas in robotics and computer vision" (NizH) and "The high compression ratio, coupled with a fast decoding time make the contribution a very appealing solution for embedded robotics downstream tasks" (SYPp).
>
> We also highlight our method's ability to generalize to shapes unseen during training (cf. Figure 7, main paper) while still maintaining high reconstruction accuracy. We have further extended our generalization experiment as part of the rebuttal (cf. A3) and show that our method can accurately reconstruct objects at different levels of data sparsity and noise. The ability to reconstruct objects in the wild is a key component of any robotic perception pipeline. In addition, as demonstrated by our AccuCities experiment, our method can also encode large-scale environments. By choosing the desired level of detail for expansion, our learned representation can be scaled at runtime for compute and memory constraints.
>
> [A] Volumetric grasping network: Real-time 6 dof grasp detection in clutter, CoRL'20
>
> [B] Dex-NeRF: Using a Neural Radiance Field to Grasp Transparent Objects, CoRL'21
>
> [C] Vision-only robot navigation in a neural radiance world, RAL'21
>
> [D] Autolabeling 3D Objects with Differentiable Rendering of SDF Shape Priors, CVPR'20
>
> [E] ShAPO: Implicit Representations for Multi-Object Shape, Appearance, and Pose Optimization, ECCV'22
>
> [F] Convolutional Occupancy Networks, ECCV'20
>
> [G] Neural Fields as Learnable Kernels for 3D Reconstruction, CVPR'22

---

### Meta-Review · Area_Chair_C8ua · 2022-08-13

**Recommendation:** Accept (Poster)
**Confidence:** 4

**Metareview:**

The paper proposes an efficient implicit recursive octree auto-decoder (iROAD) to encode large scale 3D shape datasets in real time. Overall, the reviews are positive. Reviewers agree that the approach is new and well-motivated, and the experiments are convincing.

Authors are encouraged to respond to reviewers' comments and questions. Authors should also clarify how this relates to robot learning.

Update:

Reviewers appreciate the detailed responses and additional experiments, and unanimously agree that the paper should be accepted; the work presents a solid contribution with impressive empirical results that will be beneficial to the robotics community as a strong solution for 3D vision.